# A Treatment for Rice Straw and Its Use for Mealworm (*Tenebrio molitor* L.) Feeding: Effect on Insect Performance and Diet Digestibility

**DOI:** 10.3390/insects15080631

**Published:** 2024-08-22

**Authors:** Jorge Saura-Martínez, Ana Montalbán, Jesús Manzano-Nicolás, Amaury Taboada-Rodríguez, Fuensanta Hernández, Fulgencio Marín-Iniesta

**Affiliations:** 1Group of Research Food Biotechnology-BTA, Department of Food Science, Nutrition and Bromatology, Regional Campus of International Excellence “Campus Mare Nostrum”, University of Murcia, Espinardo, 30100 Murcia, Spain; j.sauramartinez@um.es (J.S.-M.); jesus.manzano1@um.es (J.M.-N.); ataboada@um.es (A.T.-R.); 2Department of Animal Production, Faculty of Veterinary Science, Regional Campus of International Excellence “Campus Mare Nostrum”, University of Murcia, Espinardo, 30100 Murcia, Spain; ana.montalban1@um.es (A.M.); nutri@um.es (F.H.)

**Keywords:** insect larvae, agro-industrial by-products, *Myceliophthora thermophila*, laccase activity, feed conversion ratio

## Abstract

**Simple Summary:**

Mealworms (*Tenebrio molitor*) have a great ability to bioconvert vegetable by-products used as feed into body mass. Rice straw is a vegetable by-product that is difficult to dispose of and causes environmental problems. The use of physicochemical or enzymatic treatments to improve the digestibility of these by-products can increase the performance of mealworms in terms of yield. In this study, mealworm larvae were reared on diets formulated with rice straw and rice straw subjected to a combined treatment of laccase enzyme, ultrasound, and ascorbic acid to evaluate the impact on the growth performance and digestibility of *T. molitor*. The combined sonication, laccase enzyme, and ascorbic acid treatment of rice by-products facilitated their use by *T. molitor* larvae, increasing the weight gain, feed conversion ratio, and conversion efficiency of ingested feed. The use of treatments capable of improving the usefulness of vegetable by-products opens the possibility of developing new methodologies for their use for insect feed.

**Abstract:**

The development of reuse processes for plant by-products for both animal and human food offers numerous possibilities for quality-of-life improvements that align with a circular economy model. For this reason, we divided this study into two experiments. First, we designed a combined treatment consisting of laccase, ultrasound, and ascorbic acid to hydrolyze rice straw plant fibers and used the resulting feed as the basis for *T. molitor* diets. Second, we formulated diets with different inclusion levels (0%, 25%, 50%, 75%, and 100%) of rice straw and treated rice straw to assess their impact on larvae growth and diet digestibility. For each treatment, six replicates were employed: four for the growth–performance–digestibility trial and two for complementary uric acid determination tests. The combined laccase enzyme, ultrasound, and ascorbic acid treatment hydrolyzed 13.2% of the vegetable fibers. The diets containing treated rice straw resulted in higher larvae weight and a better feed conversion ratio; however, reaching 100% by-product inclusion values led to similar results between both diets. In conclusion, these treatments improve the potential of low-nutritional-value vegetable by-products as part of a *T. molitor* diet, opening the possibility of new methodologies for the use of recalcitrant vegetable by-products for insect rearing.

## 1. Introduction

Agro-industrial by-products are an important avenue in the search for new sources of nutrients with low carbon footprints for both human consumption and animal feed. This process is part of the circular economy concept, which seeks sustainability through the maximum reuse of resources [1]. There are many types of industrial by-products that are difficult to use, either due to their difficulty in terms of handling and treatment or their lack of known applications. [2,3]. Within the agri-food industry, rice production produces more by-products compared to other crops in terms of quantity. In Spain, during the 2015/16 period, 842,507 tons of rice were produced, with 14% correspond to the Valencia Community, were 75,000 to 90,000 tons of rice straw were generated in the same period [4]. The production of these vegetable by-products represents a huge source of underused resources that can be exploited in numerous ways [3,5].

The main property that makes these by-products so difficult to reuse is their vegetable fibrillar structure. The three main components of lignocellulosic biomass are cellulose, hemicellulose, and lignin [6]. Cellulose is a polysaccharide composed of a linear chain of d-glucose linked by β-(1,4) glycosidic bonds that is insoluble in water and some organic solvents. Hemicellulose is composed of branched biopolymers with different links (pentoses (β-d-xylose and α-l-arabinose), hexoses (β-d-mannose, β-d-glucose, α- and d-galactose), and/or uronic acids (α-d-glucuronic, α-d-4-O-methyl-galacturonic, and α-d-galacturonic acids) that bind between the cellulose chains, increasing their structural integrity. Finally, lignin is an aromatic polymer synthesized from phenylpropanoic precursors that is mainly linked together by a set of bonds that form a complex matrix [6]. These three polymers are found in different proportions depending on the plant species, as shown in Figure 1.

The proportions of components and their spatial structures confer different properties to each type of lignocellulosic fiber. Furthermore, recalcitrant characteristics are common to all these materials. The component proportions present in lignocellulosic fibers in rice straw are as follows: cellulose, 32–47%; hemicellulose, 19–27%; and lignin, 5–24%. Xylose and arabinose are the main pentoses [7].

This lignocellulosic structure gives these by-products a recalcitrant character that makes their application in animal feed difficult; however, with appropriate treatments, their use could be improved. To increase rice straw digestibility, treatments can be applied to degrade the indigestible fibers of the straw and break down its three-dimensional structure [6]. Different treatments have been shown to help improve the digestibility and utilization of lignocellulosic by-products, but the effect of treatments varies depending on the treatment–sample combination applied [8,9,10,11,12,13]. Thermal or chemical treatments have a good effect but generate a significant environmental impact [14]. On the other hand, enzymatic treatments lack this environmental effect but also require very specific conditions to produce the desired effect, making them difficult to apply on an industrial scale [15,16,17]. The combination of two or more biodegrading treatments can produce a synergistic effect that enhances the digestibility of these high-lignin vegetable by-products [13,14,18], as shown in Figure 2. 

Laccase is an enzyme with a delignifying capacity. Laccases (EC1.10.3.2, p-diphenol:oxygen reductase) are a family of multicopper oxidases that catalyze the oxidation of a wide range of aromatic compounds using molecular oxygen, subsequently producing water [19,20,21]. These enzymes can be found in numerous species of fungi, insects, bacteria, and plants [19,21,22]. Laccase enzymes can attack and degrade lignin and other phenolic compounds in conjunction with mediators of laccase activity (LMS) [23,24]. By biodegrading lignin, cellulose and hemicellulose become more accessible, allowing these fibers to interact with external elements [25]. The main advantage of the enzymatic treatment with laccase compared to other degradation treatments is its low environmental impact since the enzyme only uses oxygen to carry out its activity, producing water as a product of the reaction, making it environmentally friendly and waste-free [25]. The disadvantages of enzymatic treatments are that they require very specific conditions in terms of temperature and pH, among other factors, for optimal enzymatic activity [19]. Another treatment that has shown great efficacy in terms of degrading fibers with a high amount of lignin is sonication treatment [9,26,27]. Sonication breaks down the lignin in the vegetal cell wall, exposing the internal fibers and increasing bioavailability. It has been shown that the application of combined treatments can achieve greater fiber degradation since the effect of ultrasound on the plant tissue can improve the availability of the substrate for the enzyme, improving the biodegradation effect [9,28].

One animal species that has been observed to be able to feed on various plant by-products is the mealworm. The mealworm is the larval form of *Tenebrio molitor* L. (Coleoptera: Tenebrionidae), a cosmopolitan beetle that has been shown to have a powerful digestive system that allows it to survive and biodegrade high-lignin vegetal by-products [29,30]. *T. molitor* can bioconvert wastes with low nutritional content, such as polyethylene, polystyrene, or lignocellulosic by-products, into biomass [29,30,31]. This makes it a very sustainable source of protein due to its high nutritional value; in addition, its maintenance generates a lower environmental impact than other sources of animal protein, such as pork, beef, or chicken meat [32,33]. The use of lignocellulosic by-products from rice production as food for *T. molitor* for human or animal consumption could be an interesting application, using the circular economy model and increasing the sustainability of the production process [29]. 

The aim of this article is to determine the effectiveness of a combined laccase enzyme, ultrasound, and ascorbic acid treatment on the performance and digestibility of rice straw (RS) used as feed for *T. molitor* larvae.

## 2. Materials and Methods

The design of this study was divided into two experiments. The first one consisted of the combined ultrasonic, ascorbic acid, and laccase treatment of RS, and the treatment effect on the fibers was analyzed. In the second experiment, ultrasound, ascorbic acid, and laccase-treated rice straw (TRS) were included in mealworm diets, and its effect on insect performance and diet digestibility was evaluated.

### 2.1. First Experiment: RS Treatment Method

RS was obtained from the Albufera Natural Park (Valencia, Spain). Ascorbic acid (AH2), 4-tert-butylcatechol (TBC) was purchased from Sigma (Madrid, Spain). Stock solutions were prepared in 0.15 mM acetic acid to prevent auto-oxidation. Milli-Q system ultrapure water was used throughout. Laccase was derived from *Myceliophthora thermophila* (MtL, Novozym 51003, Bagsvaerd, Denmark, 1000 U/g); one unit (U) of laccase enzymatic activity on 1 mM 4-tert-butylcatechol (TBC) substrate is the quantity of enzyme that generates one micromole of product 4-tert-butyl-o-benzoquinone (TBQ) per minute in 50 mM sodium acetate buffer at 25 °C. Under these assay conditions, the molar absorptivity of TBQ at 410 nm was 1100 M^−1^ cm^−1^.

To determine the degradative effect on the fibrillar structure of the RS, three treatments and their combinations were used (enzymatic with laccase, chemical with ascorbic acid, and physical with ultrasound), obtaining 8 experimental groups (Figure 3): an untreated RS control group (CON), RS treated with ascorbic acid (ASC), RS treated with laccase enzyme (LAC), RS treated with ascorbic acid and laccase enzyme (ASC-LAC), RS treated with sonication (US), RS treated with sonication and ascorbic acid (US-ASC), RS treated with sonication and laccase enzyme (US-LAC), and RS treated with sonication, ascorbic acid, and laccase enzyme (US-ASC-LAC). Six batches of each group were made and analyzed in triplicate.

In each experimental group, RS was dried and ground using a grinder Retsch ZM200 (Retsch^®^ GmbH; Haan, Germany) at 8000× *g* to a particle size of 0.5 mm. RS (10% *w*/*v*) was immersed in an aqueous solution of 50 mM sodium acetate buffer (100 mL, pH 4.5) contained within a double-wall cylindrical vessel (4 cm internal diameter; 6.5 cm height), where water was circulated using a refrigerated bath model Digit-Cool (J.P. Selecta^®^; Abrera, Spain) to keep the temperature at 25 °C. Depending of the selected treatment, one of the following steps was performed.

For the ascorbic acid chemical treatment, a 10% *w*/*v* ascorbic acid volume was used. The treatment conditions were as follows: 25 °C for 12 h. 

For the enzymatic treatment, the RS samples were treated with laccase (1000 U/g) for 12 h at 25 °C.

For the ultrasound treatment, once immersed, the sonication treatment (24 kHz; 105 µm; 33.31 W mL^−1^; 30 min) was applied to the RS samples using an UP200H ultrasound processor with an S3 probe (Hielscher Ultrasound Technology, Teltow, Germany). The temperature of the samples during the sonication process was constantly monitored. The sonication treatment lasted for 12 h at 25 °C.

In the case of the combined treatments, the ultrasound treatment is applied first, and the ascorbic acid and laccase enzyme are incorporated later at the same time. The addition of ascorbic acid in small amounts as a coupling reagent reduces free radicals to methoxyphenols, generating a lag period that prolongs the laccase action time [22].

Finally, samples from every treatment were totally dried at 65 °C until reaching a constant weight after treatment and then stored hermetically at room temperature prior to further analysis.

#### 2.1.1. Laccase Activity

For laccase activity determination, the method reported by Manzano-Nicolás et al. [22] was followed. The laccase enzymatic activity was followed spectrophotometrically by measuring the time that the enzyme takes to consume a certain amount of ascorbic acid through its reaction with the free radicals generated by the enzyme in its reaction with the substrate. The experimental conditions were 150 mM acetate buffer at pH 4.0 at 25 °C.

#### 2.1.2. Fiber Analysis Method

The procedure used is the one described by Van Soest et al. [34]. The plant sample was treated with a neutral detergent solution to separate the elements that dissolve in the detergent (NDS) (the cell content) from the plant cell wall (NDF), composed of the insoluble fibers and compounds associated with fibers (cellulose, hemicellulose, lignin, and other minor constituents linked to the wall of plant cells, such as tannins, cutin, and silica).

Next, the sample was subjected to digestion to solubilize the hemicellulose, obtaining the acid detergent fiber (ADF). Finally, the cellulose was solubilized via acid digestion, and the acid detergent lignin (ADL) was obtained.

#### 2.1.3. Soluble Solids Content

The soluble solids content (SSC) was determined using a digital refractometer (PAL-α; ATAGO; Tokyo, Japan) and was expressed in Brix degrees (°Bx).

### 2.2. Second Experiment: Insect Rearing Method

To prepare the diets, we used breadcrumbs, brewer’s yeast, RS, and TRS using the method outlined in the first experiment. *T. molitor* were donated by the Department of Zoology and Physical Anthropology of the University of Murcia, Spain. Prior to the experiment, the larvae were fed with oat flakes supplied with a vitamin and mineral supplement. To start the rearing process, mealworms were fasted for 48 h to empty their intestines.

Based on the results of the first experiment, the RS from the US-ASC-LAC treatment was selected for diet supplementation since this treatment had a greater effect on the hydrolyzation of RS fibers.

#### 2.2.1. Insect Rearing 

The insect rearing protocol reported by Montalbán et al. [30] was adapted, including the performance–digestibility test and the uric acid test. The procedure consisted of a 28-day trial, where an initial population of 2160 mealworms was divided into 54 replicates of 40 individuals per replica (Figure 4). Each replica was weighed and homogenized until a similar weight was obtained. The average weight per larvae was 63.87 ± 0.25 mg. The larvae were placed in cylindrical plastic containers with a radius of 50 mm and a height of 80 mm. Inside these containers, 2 g of feed was deposited in addition to a water dispenser to maintain the humidity. The environmental temperature was maintained at 25 ± 3 °C with 50–60% relative humidity. 

A total of 9 diets (6 replicates per diet) were used, using breadcrumbs (energy ingredient), brewer’s yeast (protein ingredient), and RS or TRS to replace the breadcrumbs in increasing concentrations depending on the diet. In addition, a vitamin–mineral supplement and linoleic acid were added to each diet (Table 1). The diets used were a control diet composed of breadcrumbs and brewer’s yeast (CON) and 8 diets with RS or TRS replacing bread at 25, 50, 75, or 100% (RS25, RS50, RS75, and RS100 for RS and TRS25, TRS50, TRS75, and TRS100 for TRS, respectively).

The temperature was constantly monitored and maintained at 25 °C. Throughout the rearing process, both dead larvae and pupae were immediately removed to avoid cannibalism, and their weight was monitored. Six replicates were used for each diet: four replicates for a performance–digestibility test and two replicates per diet for the uric acid complementary test for diet digestibility determination [35].

#### 2.2.2. Performance and Digestibility Test

In these replicates, feed was administered ad libitum, supplementing every 7 days if necessary, with the supplemented feed being weighed. Once the test was finished, the larvae and the remaining substrate, composed of litter and frass, were separated. The larvae were weighed and killed by freezing at −20 °C. Litter and frass were also collected, weighed, and stored at −20 °C for subsequent analysis.

#### 2.2.3. Uric Acid Assay

To estimate the dietary intake and digestibility of the diets, a uric acid assay was performed to determine the uric acid concentration in the excreta. Two replicates per diet for the uric acid complementary test for diet digestibility determination were used. The uric acid concentration in the samples was used as an indicator to estimate the amount of frass from the remaining substrate during the performance and digestibility test. By determining the frass–litter ratio, the consumption and digestibility rates were measured.

To perform the assay, we used the methodology proposed by van Broekhoven et al. [35], with slight modifications. In the replicates for uric acid determination, a one-time feed of 1.5 g dry matter (DM) was offered. The larvae were weighed every 14 days, and when the weight increase ceased, we considered that all feed was consumed and that frass constituted the entire litter. The frass was collected, weighed, and stored for the determination of the uric acid concentration.

The uric acid determination method reported by Marquardt et al. [36] based on UV-Vis spectrophotometry was used as the basis for this procedure. Here, 50 mg of the sample (rejected substrate or pure frass) was extracted into 100 mL of glycine buffer solution at pH 9.3 and incubated at 40 °C for 1 h. Following this, the uric acid concentration was determined via spectrophotometry at 285 nm.

#### 2.2.4. Performance and Digestibility Calculation

From the results obtained in the *T. molitor* performance and digestibility test, we have calculated the feed conversion ratio (FCR) and the efficiency of conversion of ingested feed (ECI). FCR corresponds to the amount of feed needed (in g) to obtain a weight gain of 1 g in terms of animal production [35,37]. ECI represents the insect’s capacity to use the feed it eats for growth [38]. This parameter depends on the digestibility of the feed and the proportion of said digested feed used to maintain the mass and that is used to generate energy [38]. To calculate these parameters, the following formulas were used:FCR = Feed intake (g DM)/Weight gained (g DM), 
ECI = (Weight gained (g DM)/Feed intake (g DM)) × 100.

With the excreta data of each of the diets, we can calculate the percentage of feed that is absorbed with respect to the total amount of feed ingested by the insect. This parameter is called approximate digestibility (AD) and is determined with the following formula [38]:AD = ((DM ingested-DM excreted)/DM ingested) × 100.

### 2.3. Statistical Analysis

The normality of the data was tested using Shapiro–Wilk normality tests. One-way ANOVA was performed to analyze the effect of the treatments on RS fibrous fractions or °Brix. The means were compared using Tukey’s test. The performance and diet digestibility data were analyzed in a 2 × 5 factorial design (2 straw type, RS and TRS, and 5 inclusion levels, 0, 25, 50, 75, and 100%) using a general linear model (GLM). Linear orthogonal contrast was used to examine the RS inclusion levels. IBM SPSS Statistics version 28.0 software (IBM Corporation, Armonk, NY, USA) was used for statistical analyses. The results were expressed as the least squares mean and standard error of the mean (SEM). Differences were considered significant if *p* < 0.05.

## 3. Results

### 3.1. RS Samples Fiber Analysis

The effects of different treatments on the fibrous fraction are presented in Table 2.

Starting from NDF, including all cell wall components, we can observe that, except for the US treatment, all other treatments caused hydrolysis in plant fibers (*p* < 0.001), with the laccase enzyme treatments being especially more effective. Thus, the US-ASC-LAC treatment was the most effective, with a decreased percentage of NDF of 13.2%, followed by the LAC treatment, with 10.40%, when compared with the CON sample (*p* < 0.05). Combining an ultrasound pretreatment with laccase and ascorbic acid shows a synergistic effect that increases hydrolysis, although it was not different from the rest of the laccase rice straws treated.

As for ADF, a similar effect on fiber hydrolysis was observed (*p* < 0.001). The most effective treatment in terms of hydrolyzing vegetal fibers was US-ASC-LAC, with a decrease of 8%, followed by LAC, with a decrease of 6.4%, when compared with the CON sample (*p* < 0.05). As with NDF, the US treatment does not have a hydrolysis effect on the fibers separately, but when used as a pretreatment, it causes a synergistic effect that increases the hydrolysis effect. ASC-LAC treatment demonstrated a decrease of 5% with respect to CON straw (*p* < 0.05). 

The straw lignin content was measured in terms of ADL and a treatment effect was also noticed (*p* < 0.001). The highest lignin hydrolysis was obtained (*p* < 0.05) in the laccase treatments, alone (LAC) or combined (US-ASC-LAC), and the ASC treatment when compared with the CON sample. Within the differentiated fractions, cellulose is the fibrous fraction most affected by the treatments used (*p* < 0.001). The highest percentage of hydrolysis (7.2%) was obtained for the US-ASC-LAC treatment, followed by LAC with a loss of 5.6% and ASC-LAC with a loss of 4.7% (*p* < 0.05). The hemicellulose content rice straw was also decreased with the treatment (*p* < 0.001), except with US treatment, which was similar to CON (*p* > 0.05).

### 3.2. Soluble Solid Content

SSC is expressed in °Bx (Table 2). Brix degrees (°Bx) is a measure of dissolved solids in a liquid and is commonly used to measure the dissolved sugar content of an aqueous solution. All treatments affect the soluble solids content of RS (*p* < 0.001). The TRS with the highest concentration of soluble solids is LAC, with 1.25 °Bx (*p* < 0.05). 

### 3.3. Performance and Digestibility

The effects of the treatment and inclusion level in relation to RS on mealworm performance and diet digestibility are shown in Table 3. From all of the data obtained during the performance and digestibility test and uric acid test, we can determine whether the combined laccase, ultrasound, and ascorbic acid treatment improves RS digestibility and the exploitation of an RS diet by *T. molitor*.

Individual final weight was affected by the inclusion level and type of straw (*p* < 0.001). Thus, *T. molitor* individuals fed with TRS diets were heavier than those fed with RS diets (111.1 and 102.1 mg/larvae as an average of all diets that included TRS and RS, respectively) (*p* < 0.05). A synergistic trend was also observed in the interaction between the two variables (*p* = 0.080). At 25%, 50%, and 75%, TRS promotes higher weights (116.1, 110.3, and 105.6 mg, respectively) than their RS equivalents (98.8, 99.6, and 93.2 mg, respectively). This difference reduced as the RS percentage in the diet increased; at 100%, very similar values were found (89.7 mg for TRS100 and 85.1 mg for RS100).

As for weight gained, it was observed that it is affected by straw type in the diet and level of inclusion (*p* < 0.001). Thus, in both types of straw, a linear decrease in weight gain was observed as the inclusion level increased, and it was always higher in the TRS groups compared to the RS groups (52.3, 46.3, 41.4, and 25.8 mg for TRS25, TRS50, TRS75, and TRS100 and 35.1, 35.7, 28.9, and 21.3 for RS25, RS50, RS75, and RS100, respectively).

In the case of total intake, it was not affected by the type of RS used (*p* > 0.05); however, it was affected by the inclusion level in the diet (*p* < 0.001). Thus, as the byproduct percentage in the diet increases, intake linearly decreases (91.3, 43.6, 40.0, and 39.3 mg/larvae as an average of all diets at 25, 50, 75, and 100% inclusion levels, respectively). It was also observed that an interaction exists between the type of straw used and the inclusion level (*p* < 0.001). 

FCR is affected by both straw type (*p* < 0.01) and inclusion level (*p* < 0.001). In the case of straw type, TRS diets had a lower FCR value than RS (1.8 and 2.2 g/g as an average of all treatments that included TRS and RS, respectively), which indicates that *T. molitor* needs to ingest less TRS to gain weight than RS. Regarding inclusion levels, in both straw types, it was observed that the FCR decreased at intermediate levels with respect to the minimum and maximum levels (1.1 and 1.2 g/g as an average of all diets at 50 and 75% inclusion and 2.3 and 2.2 g/g as an average of all diets at 25% and 100%, respectively). 

Regarding ECI, the effect of straw type (*p* < 0.001) and inclusion level (*p* < 0.001) was observed. TRS diets have higher efficiency compared with RS diets (69% and 54% as an average of all treatments that included TRS and RS, respectively); therefore, treating RS increases the amount of feed converted into mass for *T. molitor*. Regarding inclusion levels, the efficiency percentage increases at intermediate inclusion levels when compared to the other values (91% and 81% as an average of all diets at 50% and 75% inclusion and 46% and 54% as an average of all diets at 25% and 100% inclusion). An interaction effect between straw type and inclusion level is also observed (*p* = 0.004). In RS diets, efficiency increased (*p* < 0.05) from 50% inclusion, while in TRS diets, efficiency increased from 25% to 75% inclusion in the diets, decreasing at 100% inclusion (*p* < 0.05).

The percentage of pupae is also affected by both straw type (*p* < 0.001) and inclusion level (*p* < 0.001). In TRS diets, an increase in the percentage of pupae was observed in comparison with RS diets (13% and 8% as an average of all treatments that included TRS and RS, respectively). The higher the inclusion level, the lower the percentage (19.5, 9.7, 4.2, and 5% as an average of all diets at 25, 50, 75, and 100% inclusion levels, respectively). As for mortality, it does not seem to be affected by the straw type used (*p* > 0.05) but by inclusion level in the diet (*p* < 0.001). Increasing the inclusion levels elevates the percentage of mortality (1, 1, 3.5, and 5.43% as an average of all diets at 25, 50, 75, and 100% inclusion levels, respectively). The interaction between straw type and inclusion level was also observed (*p* < 0.05). When using TRS diets, mortality was increased (*p* < 0.05) at higher inclusion levels (6.9% at TRS100). 

Finally, regarding AD, there is a trend relating to the effect of straw type (*p* = 0.072), and AD in TRS diets is slightly higher compared with RS diets (64.66% and 61.23% as an average of all treatments that included TRS and RS, respectively). Regarding inclusion level, in general, the approximate digestibility of the diets (AD) in terms of percentage decreased linearly as rice straw inclusion levels increased (*p* < 0.001). For RS diets, this effect was observed (*p* < 0.05) from 50% of RS inclusion (87.9, 54.6, 53.2, and 35.8% for RS25, RS50, RS75, and RS100% inclusion, respectively), and for TRS diets from 75% of TRS inclusion (*p* < 0.05) (81.9, 73.6, 49.6, and 43.4 for TRS25, TRS50, TRS75, and TRS100, respectively). For this reason, an interaction was found between both factors (*p* < 0.001).

## 4. Discussion

### 4.1. First Experiment: RS Treatment

The combined US-ASC-LAC treatment hydrolyzed RS fibers significantly more than the other assayed treatments for all fibrous fractions (*p* < 0.001). These combined treatments showed a synergistic effect that has also been observed in other studies [13,14,18]. Other assayed treatments where laccase was used (ASC-LAC and US-LAC) also showed a high degradative effect but were not as effective as US-ASC-LAC.

The laccase enzymatic treatment has proven to be the most effective single treatment assayed in terms of degrading RS fibers. Laccase is an enzyme widely used in nature by lignin-degrading organisms, including fungi such as *Trametes versicolor* or bacteria such as *Bacillus ligniphilus* [39]. This group of enzymes oxidizes phenolic compounds, such as monomers that make up plant fibers, changing their structure and depolymerizing them [19]. As observed in other studies, such as the work of Nazar et al. [39], the fibrillar structure of RS is altered by laccase, depolymerizing the plant fibers. Scanning electron microscope images show the alterations of the fiber surface that make it more porous and accessible. At the molecular level, a decrease in C, Na, and K content is observed. The decrease in C content is due to the enzyme breaking the C-C and C-H bonds, which are responsible for the bonding between lignin monomers [39]. This effect has also been observed in other lignocellulosic by-products, such as wheat chaff [40], wheat straw [41], and corncob [42].

The use of ascorbic acid has been shown to improve the effectiveness of enzymatic treatments in degrading plant fibers into other by-products [43]. Its effect consists of preventing the repolymerization of lignin fibers through a mechanism of electron transfer, facilitating hydrolysis by the enzyme and resulting in a synergistic effect between the two treatments. In the case of laccase, ascorbic acid acts as a coupling reagent, reducing free radicals to methoxyphenol and generating a lag period that prolongs the enzyme’s action time [22]. This is observed when comparing the results of the treatments, where the US-ASC-LAC treatment (49% for NDF, 30.4% for ADF, 2.7% for ADL) has a greater hydrolytic effect than US-LAC (53.6% for NDF, 33.5% for ADF, 3% for ADL) due to the synergistic effect among treatments. The most hydrolyzed fibrillar fraction is cellulose, with 2.5%. These results are like those found in other studies, such as the work of Sheng et al. [43], where an ascorbic acid pretreatment facilitated vegetable fiber digestibility in other by-products, such as wheat straw, corn stover, and corncob, improving the hydrolysis ratio by more than 10%.

Ultrasound treatments have also been shown to be effective in degrading vegetable fibers [27,44]. Energy release produced via the rupture of cavitation bubbles affects the fibrillar structure [9]. Applying ultrasound as a pretreatment facilitates laccase enzyme action in depolymerizing fibers. A synergistic effect in the combined treatments is also observed. The US-ASC-LAC treatment (49% for NDF, 30.4% for ADF, 2.7% for ADL) proved to have a greater hydrolytic effect than ASC-LAC (53.8% for NDF, 33.4% for ADF, 3.2% for ADL). The most hydrolyzed fibrillar fraction is cellulose, with 2.7%.

When comparing the SSC results of different delignifying treatments, soluble solids that were not present in the untreated straw sample were observed. In particular, when laccase enzyme treatments were used, the SSC was higher. Other authors have found that laccase depolymerizes fibers, releasing intermediate compounds and monomers [19,23]. Although the treatment with the highest degradative effect was found in the use of US-ASC-LAC, it is not the highest treatment in terms of SSC. It is possible that pretreatment with ascorbic acid and ultrasound leads to less soluble substances that are not detected within the SSC.

### 4.2. Second Experiment: Insect Rearing 

After 28 days of comparatively studying diets containing RS and TRS, it has been determined that the combined treatment (US-ASC-LAC) affected the results of the performance and digestibility tests, increasing the final individual weight of the larvae, inducing weight gain, feed conversion ratio, and efficiency, and favoring the formation of pupae. The diets used in this study are high-fiber diets; they also consist of a very lignified fiber of lower nutritional quality with a low protein content. As the amount of RS in diets increased, the protein concentration decreased. The combined treatment decreased the concentration of dietary fiber, as shown in Table 1 (24.8 g/100 g DM RS25, and 21.5 g/100 g DM TRS25, a 13.3% decrease in fiber concentration), consequently increasing fiber digestibility. The combined treatment improved the nutritional value of RS, resulting in diets with 25 to 75% TRS that lead to significantly greater weight gain per individual than in the case of RS diets. However, with 100% RS or TRS incorporation, the results are similar. For this reason, it is necessary to include other ingredients to obtain an optimal balanced diet, since both TRS and RS have poor nutritional profiles that prevent correct larval development [29,30,45]. In fact, the use of high-protein diets improves *T. molitor* rearing performance [46]; therefore, protein scarcity in diets may result in low performance at the time of mealworm development. Similar results have been observed in related studies, such as the work of Montalbán et al. [30], where increasing the proportion of protein and fiber and reducing starch resulted in reduced digestibility and increased intake, increasing weight gain. In other studies, such as the work of Baldacchino et al. [47], where diets using tomato pomace as an ingredient were used, a tomato pomace content of 41% did not have a negative effect on performance; however, at higher percentages, mealworm development time increased. This indicates that although an adequate balance of protein, fat, and fiber is not essential for mealworm development, the correct proportion of each fraction has a great influence on the effectiveness of each diet [48,49]. 

*T. molitor* has been proven capable of bioconverting a wide variety of by-products of low nutritional value [29,30]. Studies such as those conducted by Peng et al. [31] or Przemieniecki et al. [50] show that *T. molitor* is even capable of bioconverting polystyrene, a plastic polymer with practically no nutritional value. This capacity is dependent on the gut microbiota of mealworms. *T. molitor* has shown the capability to metabolize cellulose thanks to the microbiota present in its intestine [51]; however, the concentration of lignin present in some by-products makes them difficult to digest. Mealworms require an omnivorous diet at all life stages [52], which means that they are not as efficient in taking advantage of lignocellulosic by-products as other species of herbivorous insects, such as termites [53].

The effect of laccase depolymerization on lignocellulosic by-products improves the nutritional value in two different ways. First, vegetal fiber depolymerization releases oligomers that could be used by the symbiotic microbiota present in mealworms [19,23]. Second, the degradation of the plant fibrillar structure means that cellulose and hemicellulose fibers are more accessible to mealworms and microorganisms in their digestive tracts, which have cellobiose, cellulase, and amylase activity [51].

In TRS diets, an increase in the number of individuals that reached the pupal stage was observed (8% RS and 13% TRS). This may be because, by improving fiber digestibility, the larvae can obtain more nutrients, allowing them to reach this development stage earlier. This corroborates the suggested increase in soluble solids observed after US-ASC-LAC treatment in the RS samples. These enzymatic products can be used by mealworms and their intestinal microflora for development, which can lead to a greater weight gain. Similar results were observed in the study conducted by Sabarez et al. [40], where an increase in sugar concentration and other phenolic compounds was also observed after a sonication–laccase treatment was applied to wheat chaff. Other studies have shown that laccase treatment of plant by-products alters their fibrillar structure, improving digestibility and the fermentation process to produce bioethanol [54,55,56].

Although the results obtained show a positive effect on mealworm development thanks to the combined RS treatment, dietary supplementation is essential for optimal development. Further studies are needed to determine how to improve the application of low-nutritional-value by-products and their incorporation in the diet to obtain an economical and environmentally friendly insect rearing process. 

## 5. Conclusions

Of all the single treatments proposed for RS, laccase treatment is the most effective; however, when adding ultrasound pretreatment and ascorbic acid, a synergistic effect was achieved that increased its effectiveness. This combined sonication, ascorbic acid, and laccase enzyme treatment facilitates the use of RS by the larvae of *T. molitor*, increasing the weight gain, feed conversion ratio, and efficiency of conversion of ingested feed and thus increasing digestibility. Even though the most balanced diet (CON) is the one that obtains the best performance, the use of RS for insect rearing is an alternative, as it is a waste product that is not very useful for other livestock production processes. These results indicate that by using treatments capable of reducing the recalcitrant properties of RS, this vegetable by-product can be used as a feed ingredient, reducing the environmental impact at a low production cost. The improvement obtained after treatment opens a new interesting line of inquiry for the future of insect rearing for human food and animal feed, opening the possibility of developing new methodologies for the use of vegetable by-products for insect feed aligned with the circular economy model. 

## Figures and Tables

**Figure 1 insects-15-00631-f001:**
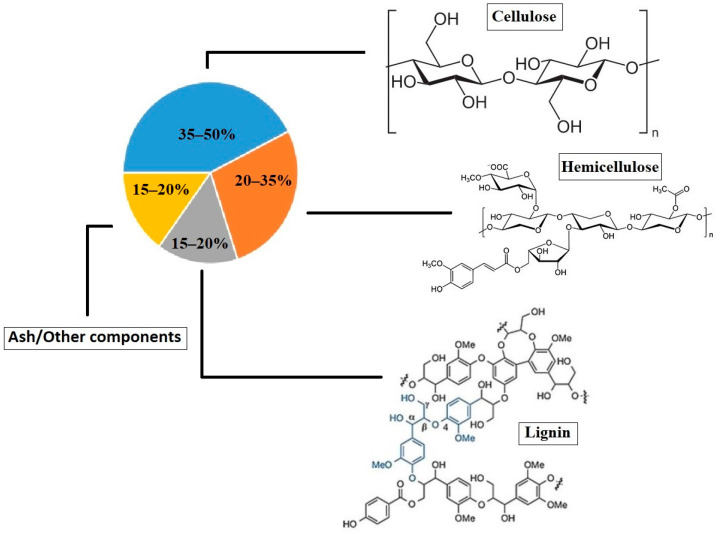
General proportions of the main components (cellulose, hemicellulose, and lignin) of lignocellulosic plant fibers.

**Figure 2 insects-15-00631-f002:**
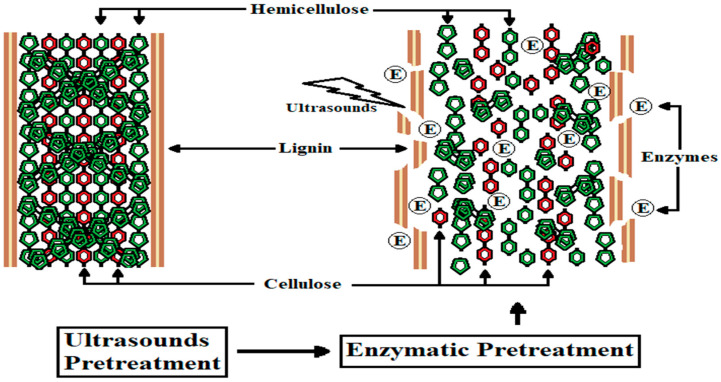
Lignocellulosic biomass pretreatment using a combined ultrasound and enzymatic process to remove lignin and degrade vegetal fibers.

**Figure 3 insects-15-00631-f003:**
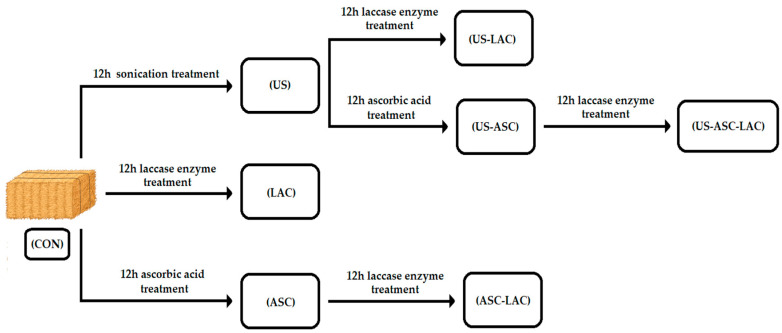
RS combined treatment experimental design. Acronyms are explained in the text above.

**Figure 4 insects-15-00631-f004:**
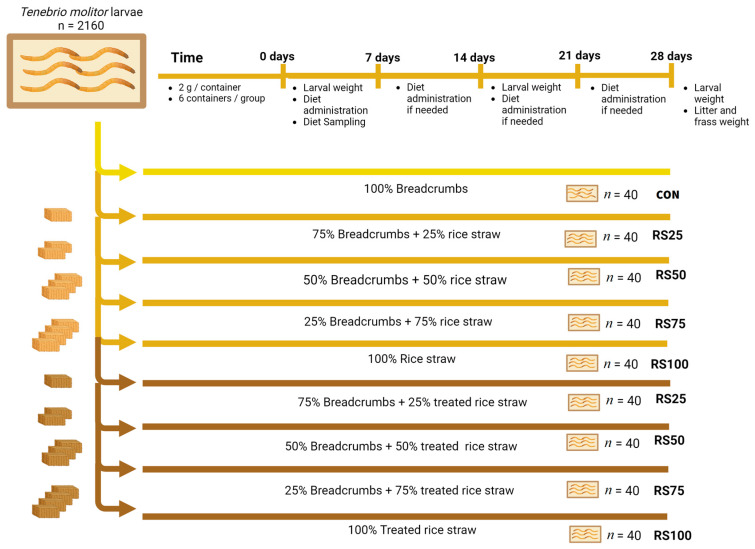
Insect rearing experimental design.

**Table 1 insects-15-00631-t001:** Composition of the experimental diets for *T. molitor*.

	Diets								
Ingredients (g/100 g DM)	CON	RS25	RS50	RS75	RS100	TRS25	TRS50	TRS75	TRS100
Breadcrumbs	86.1	61.1	36.1	11.1	0.0	61.1	36.1	11.1	0.0
Rice straw	0.0	25.0	50.0	75.0	86.1	0.0	0.0	0.0	0.0
Treated rice straw	0.0	0.0	0.0	0.0	0.0	25.0	50.0	75.0	86.1
Brewer’s yeast	10.0	10.0	10.0	10.0	10.0	10.0	10.0	10.0	10.0
Supplement ^1^	3.4	3.4	3.4	3.4	3.4	3.4	3.4	3.4	3.4
Linoleic acid	0.5	0.5	0.5	0.5	0.5	0.5	0.5	0.5	0.5
Calculated composition (g/100 g DM)									
Crude protein	16.9	14.3	11.7	9.1	8.0	14.3	11.7	9.1	8.0
Starch	67.8	48.1	28.5	8.8	0.1	48.1	28.5	8.8	0.1
Neutral detergent fiber (NDF)	13.0	24.8	36.6	48.4	53.6	21.5	30.0	38.5	42.2

^1^ The supplement provided the following per kilogram of diet: cholesterol, 5000 mg; ascorbic acid, 3000 mg; choline, 1250 mg; inositol, 250 mg; niacin, 100 mg; d-pantothenic acid, 50 mg; riboflavin, 25 mg; folic acid, 25 mg; thiamine, 25 mg; d-biotin, 1 mg; pyridoxine, 25 mg; potassium phosphate monobasic, 7440 mg; calcium carbonate, 5040 mg; tricalcium phosphate, 3576 mg; potassium chloride, 2880 mg; sodium chloride, 2520 mg; magnesium sulfate·7H**_2_**O, 2160 mg; ferric phosphate, 352 mg; sodium fluoride, 13.68 mg; copper sulfate·5H**_2_**O, 9.36 mg; manganese sulfate·H**_2_**O, 4.8 mg; potassium aluminum sulfate, 2.16 mg; potassium iodide, 1.2 mg.

**Table 2 insects-15-00631-t002:** Effect of different treatments on rice straw (RS) fibrous fraction (% DM).

	Fibrillar Composition (%)					SSC (°Brix)
Rice Straw Sample	NDF ^1^	ADF	ADL	Hemicellulose	Cellulose	
CON ^2^	62.2 ± 4.8 ^a^	38.4 ± 2.1 ^a^	3.5 ± 0.8 ^ab^	23.8 ± 2.8 ^a^	34.9 ± 2.8 ^a^	0.0 ± 0.0 ^h^
ASC	56.3 ± 4.8 ^b^	35.8 ± 2.2 ^b^	2.9 ± 0.7 ^cde^	20.6 ± 3.2 ^b^	32.8 ± 2.4 ^ab^	0.6 ± 0.0 ^e^
LAC	51.8 ± 4.9 ^bc^	32.0 ± 5.8 ^cd^	2.6 ± 0.4 ^e^	19.9 ± 3.5 ^b^	29.3 ± 2.9 ^cd^	1.3 ± 0.0 ^a^
US	62.2 ± 4.0 ^a^	39.2 ± 2.8 ^a^	3.9 ± 0.9 ^a^	23.1 ± 2.0 ^a^	35.3 ± 3.2 ^a^	0.4 ± 0.0 ^g^
US-ASC	55.6 ± 3.4 ^b^	34.9 ± 1.7 ^bc^	3.3 ± 1.1 ^bc^	20.8 ± 2.0 ^b^	31.6 ± 1.6 ^bc^	0.5 ± 0.0 ^f^
US-LAC	53.6 ± 6.6 ^bc^	33.5 ± 4.6 ^bc^	3.0 ± 0.9 ^bcde^	20.2 ± 2.3 ^b^	30.4 ± 4.8 ^bc^	1.1 ± 0.0 ^c^
ASC-LAC	53.8 ± 4.5 ^b^	33.4 ± 2.2 ^bc^	3.2 ± 1.1 ^bcd^	20.4 ± 2.6 ^b^	30.2 ± 3.1 ^cd^	1.2 ± 0.0 ^b^
US-ASC-LAC	49.0 ± 5.7 ^c^	30.4 ± 3.8 ^d^	2.7 ± 0.7 ^de^	18.6 ± 2.3 ^b^	27.7 ± 4.0 ^d^	0.9 ± 0.0 ^d^
*p*-value	<0.001	<0.001	<0.001	<0.001	<0.001	<0.001

^1^ NDF: neutral detergent fiber. ADF: acid detergent fiber. ADL: acid detergent lignin. SSC: soluble solids content. ^2^ CON, control untreated RS; ASC, RS treated with ascorbic acid; LAC, RS treated with laccase enzyme; US, RS treated with sonication; ASC-LAC, RS treated with ascorbic acid and laccase enzyme; US-ASC, RS treated with sonication and ascorbic acid; US-LAC, RS treated with sonication and laccase enzyme; US-AC-LAC, RS treated with sonication, ascorbic acid, and laccase enzyme. Means ± standard deviation values followed by different letters in the same column are different (*p* < 0.05). Standard deviations with values depicted as “±0.0” indicate values lower than 0.05.

**Table 3 insects-15-00631-t003:** Productive parameters, digestibility, and fecal uric acid levels of *T. molitor* larvae reared with rice straw- (RS) and treated rice straw (TRS)-based diets.

Item	Inclusion Level % (L)					*p*-Value ^1^			
	0 (CON)	25	50	75	100	Straw Type (T)	L	T × L	F-Value
Initial ind. weight (mg)						0.754	0.099	0.847	1.09
RS	63.5 ± 0.3	63.7 ± 0.4	63.7 ± 0.6	64.3 ± 0.3	63.8 ± 0.3				
TRS	63.5 ± 0.3	63.9 ± 0.9	64.0 ± 0.8	64.2 ± 0.5	63.6 ± 0.8				
Final ind. weight (mg)						<0.001	<0.001 ^2^	0.080	28.58
RS	133.9 ± 6.8 ^a^	98.8 ± 8.6 ^bc^	99.6 ± 7.8 ^b^	93.2 ± 3.6 ^bc^	85.1 ± 9.3 ^c^				
TRS	133.9 ± 6.8 ^a^	116.1 ± 4.5 ^b^	110.3 ± 6.3 ^b^	105.6 ± 4.6 ^b^	89.7 ± 9.8 ^c^				
Total weight gain (mg)						<0.001	<0.001 ^2^	0.100	27.38
RS	70.4 ± 7.0 ^a^	35.1 ± 8.9 ^bc^	35.7 ± 7.7 ^b^	28.9 ± 3.8 ^bc^	21.3 ± 9.2 ^c^				
TRS	70.4 ± 7.0 ^a^	52.3 ± 4.8 ^b^	46.3 ± 6.7 ^b^	41.4 ± 5.0 ^b^	25.8 ± 10.4 ^c^				
Total intake (mg/larvae)						0.236	<0.001 ^2^	<0.001	1527.9
RS	199.7 ± 1.8 ^a^	97.4 ± 2.8 ^b^	47.8 ± 4.5 ^c^	37.1 ± 1.9 ^d^	35.3 ± 5.9 ^d^				
TRS	199.7 ± 1.8 ^a^	85.3 ± 2.4 ^b^	39.5 ± 3.3 ^c^	43.1 ± 4.4 ^c^	43.3 ± 5.1 ^c^				
FCR (g/g)						0.009	<0.001 ^2^	0.057	11.17
RS	3.0 ± 0.2 ^a^	3.0 ± 0.5 ^a^	1.4 ± 0.2 ^b^	1.5 ± 0.2 ^b^	2.2 ± 1.2 ^ab^				
TRS	3.0 ± 0.2 ^a^	1.7 ± 0.1 ^bc^	0.9 ± 0.1 ^d^	1.1 ± 0.2 ^cd^	2.3 ± 0.8 ^ab^				
ECI (%)						0.001	<0.001 ^2^	0.004	14.84
RS	33.5 ± 3.0 ^b^	34.4 ± 5.9 ^b^	72.2 ± 13.1 ^a^	67.7 ± 11.8 ^a^	59.8 ± 32.6 ^ab^				
TRS	33.5 ± 3.0 ^b^	57.8 ± 4.4 ^b^	98.8 ± 14.0 ^a^	95.1 ± 19.8 ^a^	49.3 ± 16.3 ^b^				
Pupae (%)						<0.001	<0.001 ^2^	0.243	12.5
RS	14.0 ± 5.7 ^a^	16.0 ± 4.1 ^a^	5.5 ± 3.2 ^b^	2.0 ± 3.2 ^b^	2.5 ± 1.7 ^b^				
TRS	14.0 ± 5.7 ^b^	23.0 ± 5.9 ^a^	14.0 ± 4.1 ^b^	6.5 ± 2.8 ^b^	7.5 ± 3.5 ^b^				
Mortality (%)						0.747	<0.001 ^2^	0.014	4.8
RS	0.5 ± 1.1	1.0 ± 2.24	0.5 ± 1.1	6.0 ± 4.1	4.0 ± 4.5				
TRS	0.5 ± 1.1 ^b^	1.0 ± 1.3 ^b^	1.5 ± 2.2 ^b^	1.0 ± 1.3 ^b^	6.9 ± 2.0 ^a^				
Uric acid (mg/mg excreta)						0.921	<0.001 ^2^	0.583	13.73
RS	0.07 ± 0.01 ^ab^	0.08 ± 0.02 ^a^	0.05 ± 0.01 ^b^	0.03 ± 0.01 ^b^	0.02 ± 0.01 ^b^				
TRS	0.07 ± 0.01 ^a^	0.07 ± 0.01 ^a^	0.04 ± 0.01 ^b^	0.03 ± 0.01 ^b^	0.03 ± 0.01 ^b^				
AD (%)						0.072	<0.001 ^2^	0.001	36.56
RS	74.7 ± 1.9 ^b^	87.9 ± 2.8 ^a^	54.6 ± 3.1 ^c^	53.2 ± 5.3 ^c^	35.8 ± 4.0 ^d^				
TRS	74.7 ± 1.9 ^a^	81.9 ± 2.9 ^a^	73.6 ± 10.3 ^a^	49.6 ± 10.3 ^b^	43.4 ± 11.7 ^b^				

All values are presented as means ± standard deviations. Means within a row with different lowercase letters are different (*p* < 0.05). FRC = feed conversion ratio. ECI = efficiency of conversion of ingested feed. AD = approximate digestibility. ^1^ DF (n,d) = 8.36 for all variables. ^2^ In contrast analyses, a lineal effect of level inclusion was found (*p* < 0.001).

## Data Availability

The data presented in this study are available upon request from the corresponding author.

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
