# Peer review of "A Treatment for Rice Straw and Its Use for Mealworm (Tenebrio molitor L.) Feeding: Effect on Insect Performance and Diet Digestibility"

_insects, 2024, doi:10.3390/insects15080631_

Round 1

Reviewer 1 Report

Comments and Suggestions for Authors

I found the paper entitled “Treatment on rice straw and its use for mealworm (Tenebrio molitor L.) feeding: effect on insect performance and diet digestibility” particularly interesting and I believe it can be useful for all the readers who work on the same research field. In particular, after an exhaustive Introduction, comprehensive of all the important information on the topic, the M&M section provides all the information to allow other scholars to reproduce the experiments. On the contrary, although the results and discussion are informative, there are several misunderstandings provided by a not well representation of the data. This is the reason why I recommend a major revision: I believe that the paper, especially the results section and the statistical analysis, must be improved to allow a better comprehension by the readers. Following my suggestions for the improvement of the manuscript.

MAJOR ISSUES:

- Materials and Methods

I suggest to merge the materials with the methods for two reasons: 1) it is easier for the readers to read the whole protocol of an experiment together instead of reading it in different place of the manuscript; 2) by merging both sections together the authors can uniform their paper with the other papers of Insects journal.

Moreover, on how many larvae the mean and SEM value reported for the weight of the larvae is referred to (line 220)? All the 2160? Were them weighed all together before splitting them in the different replicates? For a better set up of the experiment, the mean and SEM of larval weight should be calculated after splitting the larvae into the different replicates.

In addition, it is not clear why the authors decided to split the 6 replicates, conducted for each diet, in 4 replicates to check the performance-digestibility of the larvae and in only 2 replicates for the uric acid determination. Three biological replicates should be the minimum number of replicates to consider a single experiment acceptable.

Finally, the statistical analysis was not conducted in a proper manner. The authors analysed statistical differences between different treatments by using One-Way AVOVA. This is reasonable if they only limit their analysis in the comparison of the different parameters (i.e., performance of the larvae, mortality rate, uric acid determination, and so on..) between the different inclusion levels of rice straw in the diet. Differently, they also provided in the text several comparisons between the results provided in the columns “untreated” and “treated” of the table 3. For example, from line 346 to line 347 they compared the final individual rate of the larvae derived from the two diets (untreated and treated), or at lines 368-370 they compared the FCR calculated from the insects’ performance. They also provided a p value of these analyses, indicating that their statements are not only speculative but that they were subjected to a statistical analysis. How did the authors conduct the statistical analysis on two different samples (referred to untreated and treated diets) without using a t-test? So, in addition to a proper statistical analysis, for all of these comparisons the right p-values and letters, which indicate the statistical differences, are completely missing.

In addition, it is not clear to what the SEM reported in the different tables refer to. Normally, the standard error of the mean measures how much discrepancy is likely in a sample's mean compared with the population mean. For this reason, it provides information inside a population of certain data. The authors provided a SEM related to different populations and this creates a lot of confusion in the interpretation of the data.

For all of these reasons, I suggest a strong revision of all the data from a statistical point of view.

- Results

The readability of the data represents a really big issue that affects this work. Several statements and analysis reported in the Results section lack on the basis to follow the discussion. In fact, several times throughout the text are reported data which are not present in the tables. For example, from line 349 to line 351 are reported values of TRS and RS in the text that are not reported elsewhere, making the whole discussion really difficult to follow. The same at line 353: where are these values reported? Are they an assumption or are they somehow connected with the values reported in table 3? And why in the table 3 are only reported the different inclusion levels of RS? Where are the values referred to the inclusion levels of TRS? And again, why are the values of RS reported in the text completely different whit those reported in table 3? Unfortunately, these discrepancies are many and are reported along the whole Result section, making the text really difficult to follow since almost all of these values do not support the values reported in the tables.

Moreover, it is not clear what the authors mean with “percentage of pupae” at line 389. Are these values referring to only those larvae that were able to become pupae? If yes, why only 8 and 13% were able to develop in pupae? What about the fate of the other insects? The mortality rate was really low for both treatments so it is not clear about the rest of the population.

- References

I suggest to the authors to have an in-depth look on the list of cited papers reported in the references. There are several mistakes in formatting the different papers such as i) sometimes the full name of journals is reported while sometimes it is in the abbreviated form, ii) for some papers at the end of the title a comma is reported while other times is reported a full stop, iii) genera and species must be written in italics, with only the first letter of the genera in the uppercase form.

MINOR ISSUES:

- line 15: please modify “bioconverse” with “bioconvert”.

- lines 29-30: it is not clear what the authors mean with “observe its effects”. I understand the possible effect of the treatment substrate on the larvae but in this case there are talking about the effect of the treatment. So, the statement is not clear.

- I suggest to remove the p-values from the abstract. They are not informative and make the section heavier.

- The term “by-products” sometimes is reported as “byproducts” (e.g., lines 50, 77, ..) while sometimes as “by-products” (e.g., lines 22, 52, ..). Please, uniform it as “by-products”.

- please uniform the way to provide the different headquarters since sometimes only the country is reported, sometimes city and country while other times no headquarter is indicated.

- lines 123-127: it would be maybe better to change “objective” with “aim”. Moreover, the sentence is too long and sounds not really good. I suggest to revise the entire sentence maybe subdividing it in two minor sentences.

- line 133: “its effect on insect performance and diet digestibility WAS evaluated”. The subject is “effect”.

- lines 147-154: in order to make your paper easier to read and comprehend, I suggest to provide an additional table in which this information is reported. By providing a table, I believe that all of your 8 experimental groups are easy to detect and to understand, especially when the reader moves from results to M&M section.

- line 156: please provide the x g value instead of rpm. Moreover, add a space between “0.5” and “mm”.

- lines 166 and 172: add a space between the number of the degrees and the °C. This type of error is frequently reported throughout the whole manuscript. Please check it.

- lines 175-179: the sentences here reported are not so clear. Please, revise this part for a better comprehension of the text.

- lines 182-186: the sentences here reported are not so clear. Please, revise this part for a better explanation of the text.

- lines 192-193: the sentence “It is a sequential method that shows a digestion to separate cell contents from the cell walls” is not clear. In my personal opinion this sentence can be completely removed.

- line 211: the part of the sentence “… mineral vitamin supplement to complement and fasted 48 hours before to empty their intestine” is not clear. Please make it more understandable.   

- lines from 231 to 238: maybe I didn’t understand it right but if I have a look on the total amount of the different elements of the “supplement”, it exceeds the total of 3.4 g x 100 g of diet which is reported in the table. Moreover, the comma reported in the 13.68 mg of sodium fluoride must be changed with a full stop.

- line 271: Tenebrio molitor must be written in italics.

- lines 274-275: the explanation of the ECI should be rewritten since it is not clear and creates a bit of confusion.

- line 287: please change the sentence as follow “ANOVA was performed FOR THE ANALYSIS OF..”

- line 291: please change the sentence as follow “were considered significant AT p < 0.05.” Remember that p must be written in lowercase and in italics. Please check the whole text in order to uniform it (see for example table 3, line 345, and so on..).

- the only table in which a reference is reported is the Table 1. In fact, tables 2 and 3 lack of title and brief caption. I suggest to add this information in order to improve the readability of the data. By following these modifications, the meaning of FCR and ECI below the table 3 are unnecessary since their explanations are already reported in the M&M section.

- lines 400-401: after a comparison of the values with an appropriate statistical analysis, if these DMD values will be not statistically significant the authors should modify the statement about these differences.

- at the end of line 424 a reference should be added.

- please modify the term “bioconversing” at lines 451 and 453 with “bioconvert”.

- from line 482 to line 490: the English form must be improved. It is difficult to understand the meanings of the sentences reported by the authors. Just an example: at line 484 with the terms “their scarcity in diets” I suppose the authors refer to protein but the subject of the sentence is “diets” (“high protein content diets” at line 483).

- lines 491-492: this statement is not supported by an appropriate statistical analysis (please see comments about statistics above).

- line 501: “Data shown in other studies show that..” please improve the quality of the sentence in order to avoid repetitions.

Comments on the Quality of English Language

Often the general (and specific) meaning of the sentences is not clear due to a not proper use of English Language. I believe that the revision of the English form will improve a lot the whole manuscript

Author Response

Thank you for your attention and corrections. Please see the attachment.

Reviewer 2 Report

Comments and Suggestions for Authors

The manuscript “Treatment on rice straw and its use for mealworm (Tenebrio molitor L.) feeding: effect on insect performance and diet digestibility” it is interesting because it enhances by-products rich in fibre, which are nutritionally unsuitable for breeding tenebrio. Therefore, every study that improves food efficiency represents an important increase in knowledge. I would like to point out that the manuscript has shortcomings in the writing of a scientific article; in fact the reader would appreciate a more direct flow of information and without superfluous subdivisions of paragraphs. However, much more important gaps are related to the clarity of the experimental designs: replications seem absent in the first experiment and only 4 replications are used for the second experiment (performance). The many results (and statistical analysis data) are insufficiently exposed. Control was no longer considered in the results, discussions and conclusions. Observations and suggestions (not exhaustive) have been reported below for subsequent submission.

Author Response

(The authors gave the same response as above.)

Round 2

Reviewer 1 Report

Comments and Suggestions for Authors

I appreciate the efforts the authors have been spent of the revision of the manuscript. In my personal opinion this manuscript can be accepted for pubblication. 

Author Response

Dear reviewer.

I greatly appreciate the correction work you have done on the manuscript.

Kind regards.

Mr. Jorge Saura Martínez

Reviewer 2 Report

Comments and Suggestions for Authors

Dear Authors, I appreciate the great work on improving readability. I noticed that most of the previous revisions were included in the resubmitted manuscript. However, the results section still has the same interpretation/exposition problems and the justification provided is not a solution. In this regard, I provide more details below, as well as minor observations. However, in these cases it is particularly useful for authors to subject the manuscript to reading by non-author researchers, as a test of "clarity of the results".

Specific comments:

Line 229: please, check and replace “a” with “α”;

Line 262, Table 1, Table 3 and in the text: please, standardize “C+” and “CON”;

Line 284: please, remove the space after the negative sign;

Line 285: please, add “°” at “20 C”;

Line 302: I suggest integrating this short chapter with the previous one;

Line 324: please, remove the double parenthesis;

Line 348: for consistency, add “SSC: soluble solids content”;

Line 354: please, write “p” in italics;

Line 365: the correction is not reflected in the table.

 Line 363-398: In the text, the authors continue to show differences in percentages between treatments, ignoring that these differences in most cases are not statistically significant, as shown by the results in the tables. This produces a distorted interpretation of the results. The authors should primarily highlight the significant differences between treatments and secondarily (to facilitate the reader) report some results in percentages. Furthermore, the use of p<0.001 is often ambiguous, since it seems that the p-value of the ANOVA is used to support differences between treatments (in these cases, if necessary, the p-value resulting from the post-hoc test must be used).

 Table 2: if the refractometer has a precision of 0.1% (as I think), it is not correct to report values ​​with the second decimal. Please, check and correct.

 Line 420: the values ​​reported do not agree with those in the table. These discrepancies confuse the reader.

 Table 3: the justifications for the previous revision are not at all clear. Furthermore, the table has worsened because the letters of significance are missing from the values ​​(in a previous revision I suggested the elimination of the letters only as a note on line 345, as performed, but not in the table); there are p-values ​​with superscript “2” without explanation; first column lacks header; SEM values ​​are not useful. The related text, although to a lesser extent, sometimes presents the same way of interpreting and illustrating the results (without possible confirmation in the table and with differences between theses not supported by statistical significance).

Author Response

(The authors gave the same response as above.)

Round 3

Reviewer 2 Report

Comments and Suggestions for Authors

Dear authors, I appreciate the excellent work done on improving the "Results" section and the tables. The readability of the entire manuscript is also now improved.